# Antifungal Potential of Biogenic Zinc Oxide Nanoparticles for Controlling Cercospora Leaf Spot in Mung Bean

**DOI:** 10.3390/nano15020143

**Published:** 2025-01-19

**Authors:** Zill-e-Huma Aftab, Faisal Shafiq Mirza, Tehmina Anjum, Humaira Rizwana, Waheed Akram, Muzamil Aftab, Muhammad Danish Ali, Guihua Li

**Affiliations:** 1Department of Plant Pathology, Faculty of Agricultural Sciences, University of the Punjab, Lahore 54000, Pakistan; huma.dpp@pu.edu.pk (Z.-e.-H.A.); faisalshafiq532@gmail.com (F.S.M.); anjum.dpp@pu.edu.pk (T.A.); waheedakram.fas@pu.edu.pk (W.A.); 2Department of Botany and Microbiology, College of Science, King Saud University, Riyadh 11495, Saudi Arabia; hrizwana@ksu.edu.sa; 3Department of Physics, Government Shah Hussain College, Chung, Lahore 54000, Pakistan; m.muzamilaftab@gmail.com; 4Institute of Physics Center for Science and Education, Silesian University of Technology, Krasińskiego 8A, 40-019 Katowice, Poland; 5PhD School, Silesian University of Technology, 2a Akademicka Str., 44-100 Gliwice, Poland; 6Vegetable Research Institute, Guangdong Academy of Agricultural Sciences, Guangzhou 150640, China; liguihua@gdaas.cn

**Keywords:** Cercospora leaf spot, mung bean, zinc oxide nanoparticles, antifungal property

## Abstract

Agricultural growers worldwide face significant challenges in promoting plant growth. This research introduces a green strategy utilizing nanomaterials to enhance crop production. While high concentrations of nanomaterials are known to be hazardous to plants, this study demonstrates that low doses of biologically synthesized zinc oxide nanoparticles (ZnO NPs) can serve as an effective regulatory tool to boost plant growth. These nanoparticles were produced using *Nigella sativa* seed extract and characterized through UV-Vis spectroscopy, FT-IR, X-ray diffraction, and scanning electron microscopy (SEM). The antifungal properties of ZnO NPs were evaluated against *Cercospora canescens*, the causative agent of Cercospora leaf spot in mung bean. Application of ZnO NPs significantly improved plant metrics, including shoot, root, pod, leaf, and root nodule counts, as well as plant length, fresh weight, and dry weight—all indicators of healthy growth. Moreover, low-dose ZnO NPs positively influenced enzymatic activity, physicochemical properties, and photosynthetic parameters. These findings suggest that biologically synthesized ZnO NPs offer a promising approach for enhancing crop yield and accelerating plant growth.

## 1. Introduction

Phytopathogens that cause crop losses have become a major threat to the fast-growing human population in recent decades. The agrochemicals used to control them have become ineffective because the microbes have evolved their resistance to commonly used pesticides. As a result, researchers began looking for new ways to combat microbial pathogens. With the advancement in other nanotechnology applications, the management of plant diseases using nanoparticles (NPs) is a novel strategy that has the potential to be very successful in the future. The synthesis of green nanoparticles using plant extracts is a promising field in nanotechnology, as it has economic and environmental advantages over chemical and physiological methods [1].

ZnO NPs are among the most significant nanomaterials, and their applications are growing due to their small size, peculiar forms, and intriguing physiochemical characteristics [1,2,3]. Numerous research studies have demonstrated the benefits of ZnO NPs in agricultural science and agriculture. ZnO NPs demonstrated significant antifungal potential against *Aspergillus niger* and *Candida albicans* at high doses [4]. These ZnO NPs have been demonstrated to boost photosynthetic activity in leguminous crops like soybean and mung bean [5,6]. Application of ZnO NPs to soybeans (*Glycine max* L.) has been demonstrated to have assisted in the reduction of arsenic stress [5]. Cucumber (*Cucumis sativus* L.) shoot/root biomass was boosted by using zinc-based NPs [7]. In the regions, where soils are zinc deficient, it is frequently used as a nano-fertilizer to accelerate plant growth and crop development [8]. ZnO NPs are potent antifungal agents against the pathogenic filamentous fungus *Alternaria alternata* and *Fusarium verticilliodes* [9]. ZnO NPs can be employed as a biostimulant to promote plant development and to prevent and manage phytopathogenic microorganism-caused plant diseases [10]. In a study, ZnO NPs-based antimicrobial assay was conducted against *A. niger*, *C. albicans*, *Streptococcus Aureus* and *Escherichia coli* [11].

The harmful effects of chemically created nanoparticles are a concern for researchers who are working to reduce them by using the green synthesis methodology, and it is quite efficient [12]. The process of creating nanoparticles known as “green synthesis” makes use of both plant extracts and microorganisms like bacteria and yeast [13]. In this context, there are several advantages to using plant extract-based green synthesis techniques, including their affordability, quickness, ease of use, and safety. These plant extracts produce nanoparticles of various shapes and sizes that are beneficial for a range of applications by releasing phytochemicals that function as effective capping and reducing agents. Several investigations have demonstrated that ZnO NPs may be produced from natural sources, such as plants, using green chemistry techniques [3,14,15].

The mung bean, or (*Vigna radiata* (L.) R. Wilczek var. radiata), is a short-duration grain legume mostly grown in Asia but also found in other regions of the world. It is cultivated on 7 million hectares of land globally and belongs to the subfamily Papilionaceae of the Leguminosae family. Its seed includes 24% protein and is abundant in fiber, phytonutrients, and antioxidants [16]. It is the primary summer pulse grown in Pakistan. It accounts for 85% of the total production of pulses and occupies 88% of the area in Punjab province [17]. It holds the capacity to fix atmospheric nitrogen in a symbiotic connection with Rhizobium bacteria, aiding succeeding crops while also allowing it to meet its own nitrogen needs [18].

Mung bean crop production is affected by biotic and abiotic factors. The most common diseases of mung bean are powdery mildew, bacterial leaf spot, Cercospora leaf spot, dry root rot, yellow mosaic, tan spot, and halo blight [16]. Production of mung beans is severely harmed by the serious fungal disease Cercospora leaf spot (CLS). It is a prevalent disease that can be seen in humid tropical regions of Malaysia, Bangladesh, Thailand, Indonesia, and India [19]. The CLS of mung bean is caused by the fungal pathogen *Cercospora canescens*. This disease is an important agricultural concern that affects the entire crop and, in severe cases, can cause production losses of up to 95% [20].

The medicinal plant *Nigella sativa*, which belongs to the Ranunculaceae family, is utilized worldwide because of its numerous applications. It is commonly known as black seed and has potent antimicrobial properties. Since ancient times, herbal plant extracts have been recognized to have antimicrobial properties, and tests in the early 1900s were the first to show this [21]. The use of the black seed in traditional medicine has a long and distinguished history, and the scientific literature provides us with evidence to back up the claim that it has strong antimicrobial properties against a variety of diseases. It was also more effective than many widely used anti-microbial remedies. This study investigates the synthesis and characterization of zinc oxide nanoparticles (ZnO NPs) derived from *N. sativa*, focusing on their antimicrobial activity against *C. canescens*. It examines the correlation between the physicochemical properties of ZnO NPs and their antifungal effects, as well as their potential to enhance plant growth. The study aims to evaluate the use of green-synthesized ZnO NPs as a sustainable, eco-friendly alternative to chemical fungicides in agricultural practices.

## 2. Materials and Methods

### 2.1. Collection of Plant Material

*N. sativa* seeds were acquired from a local bazaar in Lahore. The mung bean variety (NM-2011) was procured from the Nuclear Institute for Agriculture and Biology (NIAB), Faisalabad.

### 2.2. Pathogen Isolation and Confirmation of Pathogenicity

#### 2.2.1. Pathogen Isolation and Identification

The Ayub Agricultural Research Institute (AARI) in Faisalabad was surveyed for the collection of disease samples of Cercospora leaf spot disease in 2021. The standard sodium hypochlorite method was used to surface-sterilize the infected leaf samples. Infected leaves were cut into little pieces and inoculated on standard potato dextrose agar (PDA) medium with streptomycin as an antibacterial agent and the inoculated plates were incubated at 28 ± 2 °C. Obtained fungal cultures were subjected to subculturing on new media plates for purification. The pathogenic organism was initially identified using information on its morphology and colony characteristics.

The fungal isolate was then identified for molecular investigation using the internal transcribed spacer region of the ribosomal RNA (ITS rRNA). The standard CTAB technique was used to extract genomic DNA from the freshly grown fungal mycelial mass. By using the primers ITS 1 and ITS 4, the ITS region was amplified as reported by Mohamed et al. [22].

#### 2.2.2. Detached Leaf Method

To confirm the virulence of each fungal culture, Koch’s postulates were employed. Isolated fungal cultures were inoculated on potato dextrose broth when they were seven days old, and the cultures were then kept in a rotary shaker for seven to ten days. The spore suspension was serially diluted using a hemocytometer to create an inoculum with 10^6^ spores/mL. Twelve young, healthy leaves from the end of the petiole of mung bean plants were cut off and cleaned with distilled water. Six petri dishes of these leaves were used, and both lids of each plate were covered with sterile blotting paper. These leaves were inoculated using a micropipette and a 5 µL spore suspension under aseptic conditions on the upper surfaces. The remaining six healthy leaves were given a double-distilled autoclaved water treatment to serve as the un-inoculated control. Every petri dish was covered, kept at 28 °C, and regularly checked for the onset of disease symptoms. Koch’s postulates on the pathogenicity of the pathogen were fulfilled by re-isolating it from the artificially infected leaves after the disease had already been established.

### 2.3. Synthesis of ZnO NPs Using Green Approach

*N. Sativa* seeds (30 g) were cleaned, ground, and boiled for 15 min in 150 mL of double-distilled deionized water. After cooling, the Whatman No. 1 filter was used to remove impurities from the aqueous seed extract. For later use, the extract was stored in a sterilized container at 4 °C. Accordingly, 40 mL of water was used for 11.5 g of zinc sulfate. The solution was thoroughly stirred using a magnetic stirrer. A sterilized needle was used to slowly add 10 mL of *N. sativa* seed extract to this solution and stir for 10 min. The solution was then given a dropwise addition of NaOH solution (2 M). After that, it was centrifuged for 15 min at 3000 rpm. Following centrifugation, a hot air oven was used to dry the pellet and the supernatant was discarded. The predicted tiny ZnO NPs were created by grinding the powder in a mortar and pestle. For further visual and compositional characterizations, ZnO NPs were shifted and stored in a sterile container [23].

### 2.4. Characterization of ZnO NPs Synthesized Using N. sativa

Ultraviolet-visible (UV-visible) spectra were recorded between 220 to 750 nm wavelength using a DeNovix DS-C spectrophotometer (Wilmington, DE, USA) to examine the NPs absorbance and to find the absorbance changes with the modified reaction conditions. The X-ray diffraction technique was adapted by using the Rigaku 600 Miniflex system (Tokyo, Japan) to evaluate the properties and nature of the nanoparticles by using Cu k-ray (λ = 1.5412) radiation between the scan range of 100 to 800. The FTIR spectrum of ZnO NPs was captured using the Agilent Technologies instrument Cary 630 (Santa Clara, CA, USA). FTIR spectra in the range of 4000 to 450 cm^−1^ were recorded for the samples to recognize the distinct functional groups on the surface of the nanoparticles.

### 2.5. Antifungal Potential of ZnO NPs

An in vitro test was performed using an agar dilution method to evaluate the potential of synthetic nanoparticles to control *C. canescens* [24]. The PDA media was first autoclaved and then a certain quantity of NPs was added when the temperature of the medium reached 40 °C. Using a weighing device, the nanoparticles were weighed for specific concentrations of 300, 600, 900, 1200, and 1500 mg/L and added to the PDA medium and properly mixed before being poured. The medium in the culture plates was inoculated with a 0.5 cm fungal plug from cultures that were 7 days old. The control plate did not contain nanoparticles. It was inoculated with a fungal plug and grown at 25 ± 2 °C until the fungus completely covered the control petri dish. For the evaluation of antifungal activity, the diameter of the *C. canescens* colony was measured every day and the formula shown below was used to calculate the suppression of fungal growth:$$Inhibition\left(\%\right)=\frac{dc-dt}{dc}\times 100$$
where dc represents the growth diameter of the control, and dt represents the growth diameter of the treatment. The whole experiment was carried out in three independent replicates for each treatment and control to ensure reliability and reproducibility.

### 2.6. Potential of ZnO NPs to Suppress Cercospora Leaf Spot in Mung Bean

#### 2.6.1. Application of ZnO NPs in Pot Trials

The concentration of nanoparticles with potential antifungal activity during in vitro assays was further evaluated in pot experiments for its efficacy in the field. Pot experiments were conducted in the field at the Department of Plant Pathology, University of the Punjab, Lahore, Pakistan. Formalin was used to carry out soil fumigation, and 10–12 kg of soil in each 12-inch soil pot contained 0.264 g of urea, 0.600 g of triple super phosphate, and 0.520 g of potash, corresponding to 40–60–40 kg of N, P, and K per hectare; half of the fertilizer was applied at sowing and half at the vegetative phase. The mung bean seed variety used in the pot experiment was NM-2011. In each pot, 5 holes were drilled to a depth of 1 cm, and 3 seeds were planted in each one. The pathogen was grown on PDA medium plates and incubated at 25 °C for seven days. After one week of incubation, the spores were harvested in distilled autoclaved water. The inoculum was prepared as an aqueous spore suspension at a concentration adjusted to 3 × 10^5^ conidia mL^−1^. The allotted plants were inoculated by spraying with aqueous spore suspension on the leaves until run-off. Details of treatment are provided in Table 1. The control treatment was sprayed with distilled sterilized water. The solution of ZnO NPs was prepared by dispersing ZnO NPs in water using ultrasound. Pots were watered weekly with 50 mL of ZnO NPs aqueous solution and monitored for growth and germination. Experiments were performed in three independent replicates to ensure the consistency and reliability of results.

#### 2.6.2. Determination of Disease Severity Parameters

The disease severity for the plant disease index was assessed using a scale of 0 to 9 after 60 days of inoculation (Appendix A). The disease index percentage (PDI) was evaluated using the below formula.$$PDI=\frac{Sum\;of\;all\;disease\;severity\;ratings}{Total\;obervations\times Highest\;rating\;in\;scale}\times 100$$

### 2.7. Effect of ZnO NPs on the Growth and Physiology of Mung Bean Plants

#### 2.7.1. Analysis of the Plant Growth Parameters

Growth parameters were measured by following standard cultural practices. The following parameters were recorded for every treatment individually: shoot and root length, dry and fresh biomass of shoots, dry and fresh biomass of roots, root nodules, leaves number, pods per plant, pod size, and dry and fresh biomass of pods.

#### 2.7.2. Quantification of Total Chlorophyll (Chl) and Carotenoid Contents

To determine the Chl concentration, 0.05 g of ice-cold leaf samples were extracted for 24 h in dark conditions with 10 mL of 80% acetone [25]. By using a spectrophotometer (Agilent Technologies Cary 60 UV-vis) the amounts of Chl *a*, Chl *b*, and total Chl (mg/L) in the supernatant were measured. Three plant samples were selected from each treatment for evaluation. Following Alam et al.’s [26] protocol, with a few minor modifications, carotenoids were assessed. First, 40 mL of acetone, 60 mL of n-hexane, and 0.1 g of MgCO_3_ were mixed to homogenize 2 g of dried material for 5 min. Then, the mixture was micro-filtered. The filtrate was then mixed two times in 25 mL of acetone and one time in 25 mL of n-hexane. The final sample was transferred to a separatory funnel and five times washed with distilled water (100 mL). The final absorbance was measured at 436 nm using a spectrophotometer. Using the β-carotene reference curve as a standard, the absorbance levels were then measured.

### 2.8. Statistical Analysis

All the collected data were statistically evaluated using the DSASTAT [27] excel extension by performing a one-way analysis of variance (ANOVA) and Tukey’s test.

## 3. Results

### 3.1. Pathogen Isolation and Confirmation of Pathogenicity

The fungal pathogen that produces leaf spots was isolated and purified from infected leaves obtained during field research on *V. radiata*. *C. canescens* was identified as an isolated fungal pathogen based on its morphological traits. The mycelium is opaque, with irregular branches. Conidiophores are pale to olive brown, straight or slightly curved, geniculate, unbranched, and cylindrical, with zero to eight septa and five to seventeen septa. Conidia that are hyaline, obclavate-cylindric, straight to slightly curved, and have one–fourteen septa and an average diameter of 102.8–3 µm (Figure 1B,C). ITS regions were amplified and sequenced to identify fungal isolates for molecular characterization. Homology analysis revealed >99% similarity to previously submitted *C. canescens* isolates (MN795697.1), (MN795657.1), and (MN795694.1) (Appendix A).

Identification of the pathogen led to the use of test pathogens and hosts to confirm Koch’s hypothesis of pathogenicity. Two to three days after inoculation, leaves of mung bean plants developed characteristic leaf spot symptoms which were first observed during the field survey, as shown in Figure 1A. Initially, very small brown spots first appear on the leaves, which indicate the beginning of chlorosis. Gradually, the necrosis begins and the entire leaf dies (Appendix A). The etiology of the disease was verified by the isolation of *C. canescens* from the inoculated leaf tissue on which the disease rapidly progressed over time, which suggested that *C. canescens* is a virulent mung bean pathogen (Appendix A).

### 3.2. Characterization of Green-Synthesized ZnO NPs

#### 3.2.1. UV-Visible Spectroscopy

The ultraviolet-visible spectral area of the absorption spectrum for green-synthesized ZnO NPs is shown in Figure 2A. At 369 nm, there is a very strong and distinct peak that corresponds to the production of ZnO NPs. To obtain the band gap, a plot of (αhυ)^2^ vs. hυ is shown in Figure 2B. The obtained value of the band gap for ZnO was measured to be 3.12 eV, which shows excellent accord with the values reported in the literature [28,29].

#### 3.2.2. X-Ray Diffraction

The synthesized ZnO NPs were created using *N. sativa* seed extract, and Figure 2C shows the X-ray diffraction spectrum for these particles. The ZnO wurtzite (hexagonal) structure of the nanoparticles can be used to index them (JCPDS card no. 36-1451) [30]. One can see the polycrystalline nature of nanoparticles, with the strong peaks of diffraction at the 2*θ* values of 31.713°, 34.401°, 36.216°, 47.533°, 56.556°, 62.796°, 67.938°, and 68.984° corresponding to the crystallographic planes of (100), (002), (101), (102), (110), (103), (112) and (201), respectively. There is only one peak, ZnO, which indicates that there are no second-phase particles and points to the excellent quality of ZnO NPs. Appendix A contains the calculated values for crystallite size.

#### 3.2.3. Fourier-Transform Infrared Spectroscopy (FTIR)

Figure 2D displays the absorbance bands of *N. sativa*-derived ZnO NPs that were produced utilizing green synthesis. Using an attenuated total reflectance (ATR) setup, the spectra were captured throughout the 1800–600 cm^−1^ range. Here we are showing only the fingerprint region, and the functional group region is omitted for the sake of clearance. The different absorption bands were detected at 618.8, 667.10, 748.7, 1198.3, 1288.6, 1399.4, 1537.2, 1556.3, 1638.8, 1652.9, and 1720 cm^−1^. The bands at 618.8 cm^−1^ and 748.7 cm^−1^ correspond to an alkane and, supposedly, a C–H band in alkynes [31] and the band at 667.10 cm^−1^ indicates the presence of sp^2^ C–H bending [23]. The 1198.3 cm^−1^ band represents the C–N stretch of the aliphatic amine peak [32]. The bands at 1288.6 cm^−1^ and 1556.3 cm^−1^ reflect the stretching of C–N and N–H bonds depicting the presence of an aromatic amine phytochemical [33,34]. The 1399.4 cm^−1^ band represents the C–H stretching vibration of the alkene group [34]. The 1537.2 cm^−1^ band depicts the presence of a bending vibration of N–H in secondary amines of proteins [30]. The 1638.8 cm^−1^ band represents the C=O stretching vibration [23,34]. The 1652.9 cm^−1^ band indicates the presence of a C=O bond of unsaturated aldehyde. Similarly, the 1720 cm^−1^ band represents the asymmetric stretching vibrations of C=O and C–H bonds [32].

#### 3.2.4. Scanning Electron Microscopy

The surface morphology of the nanomaterials was investigated using scanning electron microscopy (SEM). The obtained SEM images for different magnifications are shown in Figure 3. Most of the nanoparticles are somewhat rectangular in shape and are produced between 50 and 70 nm in diameter. The results of SEM analysis support the claim that the green extract is an important advancement as a reducing agent in the synthesis of ZnO NPs.

### 3.3. Analysis of the Antifungal Potential of ZnO NPs

ZnO NPs synthesized from *N. sativa* seed extract were observed for their antifungal activity against *C. canescens* at concentrations of (300/600/900/1200/1500 mg/L) (Figure 4). The minimal antifungal effect of ZnO NPs was observed at 300 mg/L, which exhibited 56.36% control while the maximum control was found at 1200 mg/L, exhibiting the significant effect of 84.14% when compared to the control plates. Our findings further demonstrate that the green-synthesized ZnO NPs showed antifungal activity which was dose-dependent, as illustrated in Figure 5A, where the zones of inhibition were bigger with increasing ZnO NPs concentrations except for the 1500 mg/L concentration. These findings are consistent with past research.

### 3.4. Potential of ZnO NPs to Suppress Cercospora Leaf Spot in Mung Bean

According to this study, the susceptibility of mung bean to disease reactions ranged from being moderately resistant to being extremely vulnerable. The negative control had a disease index average of 4.16% while 83.33% was observed in the positive control. Figure 5B shows that of the various concentrations tested in the pot experiment under natural conditions, the 1200 ppm concentration gave resistance (76.7%) against Cercospora leaf spot disease (CLS), whereas the plants at 900 ppm concentration were moderately (70%) resistant to the disease.

### 3.5. Effect of ZnO NPs on the Growth and Physiology of Mung Bean Plants

#### 3.5.1. Effect of ZnO NPs on the Growth of Mung Bean Plants

The beneficial impacts of ZnO NPs on a few agronomic characteristics of mung bean plants were also researched in a greenhouse environment. The use of ZnO NPs significantly improved the development and yield traits of mung bean plants in the presence as well as the absence of fungal pathogens when compared to the corresponding control plants (Table 2). Plants treated with ZnO NPs (1200 ppm) had significantly higher agronomic characteristics than non-treated control plants, including shoot length (12.5%), root length (30.5%), number of leaves (24.2%), shoot fresh weight (8.6%), shoot dry weight (43.4%), root fresh weight (13.2%), and root dry weight (20.8%) (Table 3). ZnO NPs had a favorable impact on mung bean growth parameters when applied to plants with the fungal disease *C. canescens* (Table 2). Figure 6 shows the effects of various ZnO NP concentrations on mung bean plants.

Data showed significant treatment impacts for the yield-related factors (Table 2). Plants exposed to ZnO NPs at 1200 mg/L significantly yielded a greater number of pods per plant (37.5%), pod size (12.2%), number of seeds per pod (37.5%), pod fresh weight (26.9%), and pod dry weight (27.4%) when compared to negative control plants. Effects of ZnO NPs on pod size can be observed in Appendix A.

On root nodulation, ZnO NPs undoubtedly had an effect. ZnO NPs treatment prompted plants to produce more root nodules on average. Specifically, a higher dose of 1200 mg/L considerably increased the number of nodules by 19% (Table 2). In Appendix A, root nodules appear to be evident. Similar results about the advantageous impacts of ZnO NPs on the agronomic factors of mung bean plants in the presence of the leaf spot pathogen were also noted, so they are not discussed here.

#### 3.5.2. Effect of ZnO NPs on the Physiology of Mung Bean Plants

Table 3 shows that the mung bean crop responded positively to the treatment of ZnO NPs in terms of chlorophyll content in inoculated and non-inoculated plants. The chlorophyll *a* level of mung bean plants was always higher than the chlorophyll *b* content. In the case of non-inoculated plants, at 1200 mg/L concentration, the amount of chlorophyll *a* increased by 27.9%, chlorophyll *b* increased by 28%, and total chlorophyll increased by 29.4% when compared to the negative control. In the case of inoculated plants, the second concentration (1200 mg/L) generated a considerable rise in chlorophyll levels of mung bean, increasing chlorophyll *a* content by 28.1%, chlorophyll b by 35.2%, and total chlorophyll by 32.6% (Table 3).

There were significant differences in total carotenoid content between non-inoculated and inoculated plants in mung bean plants at various concentrations. The lowest amount of carotenoid content was observed in the positive control (11.30 mg/g), while non-inoculated plants increased their carotenoid levels by 11.07% when ZnO NPs were added at a dosage of 1200 mg/L (Table 3).

## 4. Discussion

This research aimed to create a novel strategy against the fungal pathogen *C. canescens*, which causes mung bean Cercospora leaf spot. Recent advances in nanotechnology and nanoscience are making it possible to manage plant pathogen-caused diseases. In this study, we attempted to synthesize ZnO NPs using *N. sativa* seed extract using a green synthesis method. The effectiveness of the green-synthesized nanoparticles was tested in vitro and in vivo. The possibility of using herbs to create nanoparticles is an attractive aspect of sustainable nanotechnology.

The available evidence suggests that *N. sativa* and its active components, such as thymoquinone, have antifungal activity against plant pathogens and may have potential as natural alternatives to synthetic fungicides [35]. Thymoquinone, a major bioactive compound in *N. sativa*, has been extensively studied for its various medicinal properties. This phytochemical, which belongs to the quinone class, is a volatile oil that makes up about 30% to 48% of the volatile oil content of *N. sativa* seeds [36]. Known for its diverse therapeutic effects, thymoquinone demonstrates antifungal, antibacterial, antiviral, and anti-inflammatory properties [37]. These pharmacological actions are believed to result from its ability to modulate various signaling pathways and transcription factors.

Research shows that while high concentrations of ZnO NPs can exhibit phytotoxic effects, low concentrations have the potential to enhance plant growth [38]. For instance, concentrations ranging from 2000 to 4000 mg/L have been reported to induce genotoxic effects during the early growth stages of edible plants like buckwheat [39]. The narrow optimal concentration range of zinc for plant health underscores the critical need for precise dosage and controlled application [40]. In our study, ZnO NP concentrations were carefully selected within this low-dose range, ensuring no observable signs of toxicity. This approach was designed to maximize the beneficial effects of ZnO NPs on plant growth while minimizing potential adverse impacts, thereby aligning with sustainable agricultural practices.

One study published by Aftab et al. [41] investigated the antifungal activity of the methanolic extract of *N. sativa* against *Fusarium oxysporum* and *Macrophomina phaseolina*. He concluded a significant 21% inhibition in the biomass of these species and that it could be used as a natural fungicide. Another study found that thymoquinone had a potent antifungal effect against a range of fungal pathogens, including *C. albicans*, *Cryptococcus albidus*, and *Aspergillus fumigatus* [42,43]. The above-mentioned research indicates that *N. sativa* seed extract, which contains thymoquinone, has exhibited a strong antifungal effect and can be used for the green synthesis of nanoparticles.

Nanotechnology-based treatment of plant infections is becoming increasingly common due to the antimicrobial properties of nanoparticles. The antifungal activity of ZnO NPs against *C. canescens* received a few investigations. It is incredibly intriguing to work with metal nanoparticles and their oxides. A lot of research has been carried out on the biological impacts of zinc and its oxides. Depending on their size, nanoparticles can have antimicrobial properties. The antifungal ability of ZnO NPs against fungal pathogens such as *A. niger*, *Rhizopus stolonifer*, *Penicillium expansum*, *Botrytis cinerea*, and *Fusarium* spp. has been shown in a variety of research [44,45,46]. ZnO NPs distorted fungal hyphae, which prevented *B. cinerea* from proliferating by interfering with cellular functions. However, the presence of ZnO NPs prevented the conidiophores and conidia of *P. expansum* from forming, ultimately leading to the death of the hyphae [45].

Similar research conducted by [47] found that a dose of 1200 µg/mL may suppress *C. canescens* growth by up to 89% while also increasing mung bean growth and physiological parameters by 30% in an in planta experiment. In a research work to evaluate the physicochemical and antimicrobial efficacy of zinc oxide nanoparticles (ZnO NPs) created from the cell-free supernatant of *Phaeodactylum tricornutum* cultures, sweet cherries were sprayed to investigate quality retention and showed large-scale inhibition of *C. albicans*, *S. aureus*, and *E. coli* at the lowest inhibitory concentration from 0.078 to 0.156 mg/mL. In addition, 100–1000 mg/L ZnO NPs can efficiently inhibit *Mucor heimalis*. These findings imply that one of the safest and most efficient ways to protect commercial fruits for long-term preservation is through the use of green coatings based on ZnO NP [48].

ZnO NPs showed very good fungicidal activity against grapefruit fruit rot pathogens, and the most significant decrease in the growth of fungi was seen in vitro and in vivo at a concentration of 1000 g/mL of green NPs. These findings demonstrate the capability of *Trachyspermum ammi* seed extract to significantly reduce and stabilize ZnO NPs [49]. Kamal et al. [50] found that ZnO NPs had substantial antifungal activity at a number of concentrations, including 250 and 1000 g/mL. However, 0.25 mg/mL (or 88%) of ZnO NPs showed the greatest growth inhibition, whereas 0.1 mg/mL (or 22%) showed the least growth inhibition. Pachaiappan et al. [23] synthesized ZnO NPs using *Justicia adhatoda* leaves and three zinc precursors and found good antimicrobial activity against various fungal and bacterial strains. It was discovered that all of the synthesized nanoparticles restricted microbial proliferation through the production of reactive oxygen species and the release of zinc ions, both of which kill microbial organisms.

Abdelaziz et al. [51] biosynthesized ZnO NPs using *Penicillium expansum* and the in vitro antifungal activity was checked against *Fusarium oxysporum*. ZnO NPs were used to bring about a 75% reduction in the severity of Fusarium wilt to growing *Solanum melongena* L. plants. He also observed a significant rise in plant height (152.5%), root length (106.6%), fresh biomass (146%), chlorophyll *a* (102.8%), chlorophyll *b* (67.86%), total soluble carbohydrates (48.5%), total soluble proteins (81.8%), phenols (10.5%), antioxidant activity, and isozymes in comparison to control plants. In another published study, bulk ZnO NPs were synthesized and about 170 g of nanoparticles were obtained. During in vitro antifungal studies, high levels of green-fabricated ZnO NPs greatly slowed the development of isolated *Colletotrichum* sp. (KUFC 021). Additionally, high doses of synthetic ZnO NPs also substantially decreased anthracnose symptoms on orchid leaves inoculated with the *Colletotrichum* sp. (KUFC 021) under greenhouse conditions [52].

Zinc oxide nanoparticles have been extensively studied for their potential applications in agriculture, including as a means of enhancing plant growth and development. Studies have reported that ZnO NPS can enhance the development and growth of plants. According to published research, a 1500 ppm concentration of ZnO NPs produced the highest germination percentage (80%) and seedling vigor index. The yield was 42% higher than the control and 15% higher than 2000 ppm [53]. Similar studies showed that seed germination and other agronomic and physiological parameters were improved when ZnO NPs were treated at 1000 ppm concentration [54]. The effects of ZnO NPs exposure on the physiological parameters of plants can vary depending on exposure dose, exposure time, and plant species. These effects can be observed because of the difference in the concentration used or the plant species used for the evaluation of physiological parameters.

Khan et al. [55] reported that exposure of maize plants to ZnO NPs caused a substantial rise in the dry biomass, protein contents, and other physiological and enzymatic parameters of plants. The authors suggested that this positive effect may be due to the ability of ZnO NPs to enhance nutrient uptake and photosynthesis. The available evidence suggests that the ZnO NPs can be beneficial for enhancing plant growth and physiology and are dependent on a range of factors, including the concentration of ZnO NPs, the duration of exposure, and the physiological status of the plant. More research is needed to completely understand how ZnO NPs affect plant growth.

The beneficial impacts of ZnO NPs may be due to the ability of ZnO to enhance nutrient uptake and photosynthesis. The increase in chlorophyll and carotenoid content could be explained by the production of reactive oxygen species (ROS) triggered by ZnO NPs, which can act as signaling molecules and enhance plant defense mechanisms [56]. Additionally, another study published by Kareem et al. [57] reported that the application of ZnO NPs boosted the chlorophyll content and photosynthetic rate of mung bean plants. The study suggested that this positive effect on photosynthesis was due to the increase in stomatal conductance and intercellular CO_2_ concentration in the leaves, leading to improved gas exchange and ultimately resulting in increased chlorophyll and carotenoid content.

## 5. Conclusions

The findings of this study indicate that ZnO NPs have the potential to enhance both the antifungal activity and physiological performance of mung bean plants, leading to improved growth, nutritional quality, and yield. However, it is crucial to consider the concentration and exposure duration, as higher concentrations or prolonged exposure to ZnO NPs may result in negative effects on plant health. These results highlight the importance of optimizing ZnO NP application for effective pathogen control and plant growth promotion in agricultural systems.

## Figures and Tables

**Figure 1 nanomaterials-15-00143-f001:**
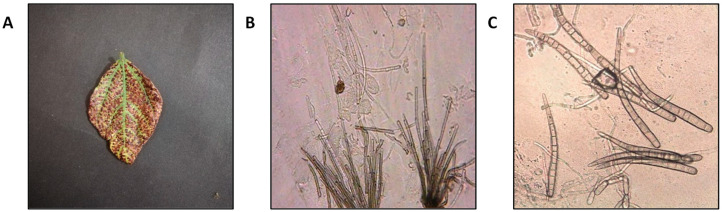
Symptoms caused by *C. canescens* (**A**); microscopic characteristics of *C. canescens* under microscope (**B**,**C**).

**Figure 2 nanomaterials-15-00143-f002:**
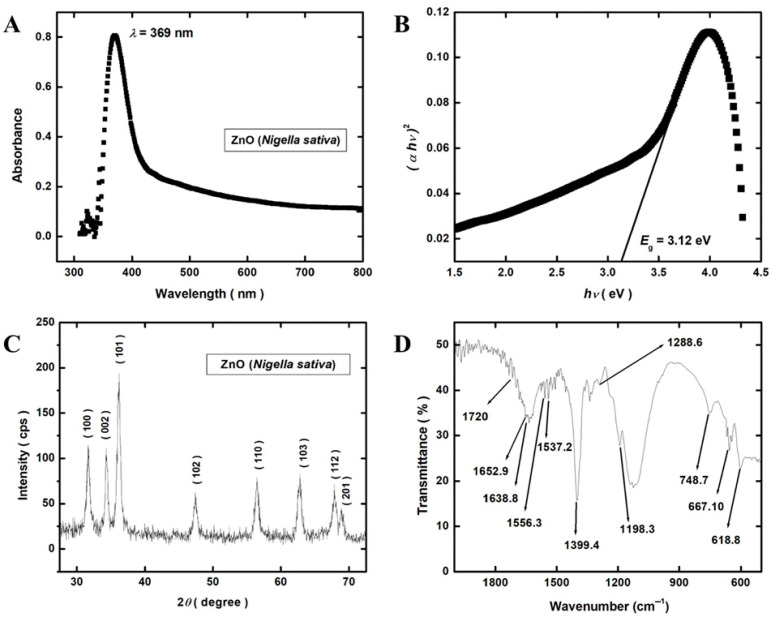
Physiochemical characterization of ZnO NPs. (**A**) UV-vis absorption spectrum of green-synthesized ZnO NPs; (**B**) band gap energy of green-synthesized ZnO NPs; (**C**) XRD pattern of green-synthesized ZnO NPs; (**D**) FTIR spectra of green-synthesized ZnO NPs.

**Figure 3 nanomaterials-15-00143-f003:**
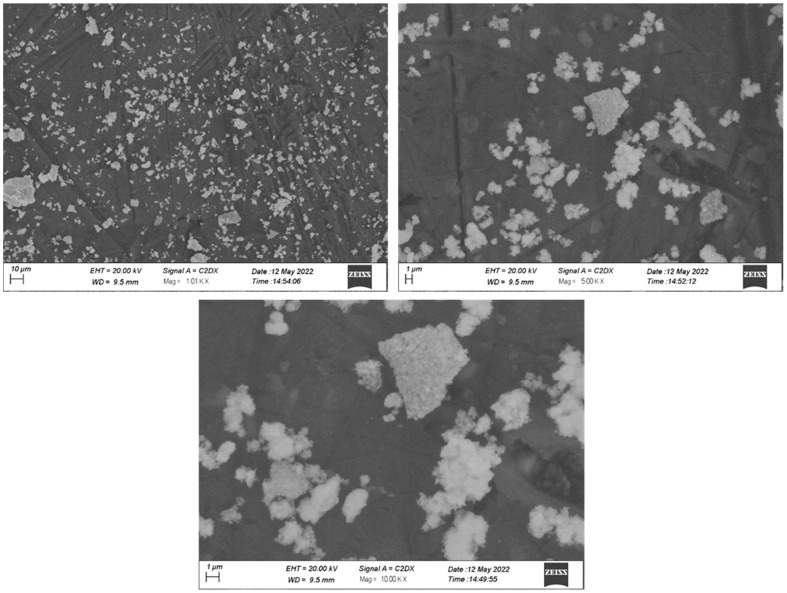
SEM images of ZnO NPs at 1000×, 5000× and 10,000× magnifications respectively.

**Figure 4 nanomaterials-15-00143-f004:**
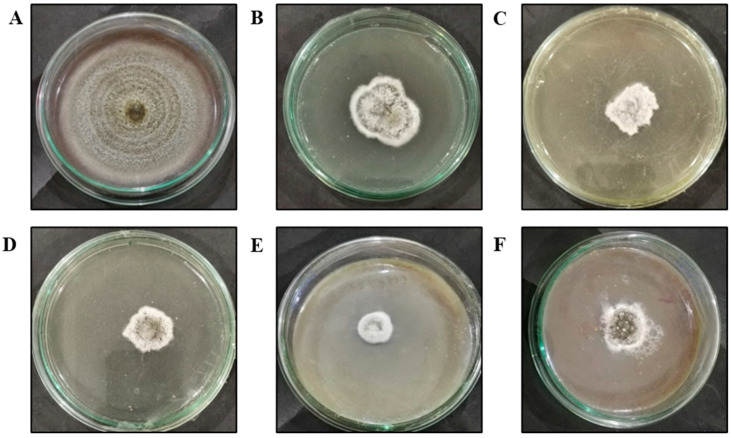
In vitro effect of ZnO NPs on antifungal activity against *C. canescens* at different concentrations: (**A**) 0 mg/L (control); (**B**) 300 mg/L; (**C**) 600 mg/L; (**D**) 900 mg/L; (**E**) 1200 mg/L; (**F**) 1500 mg/L.

**Figure 5 nanomaterials-15-00143-f005:**
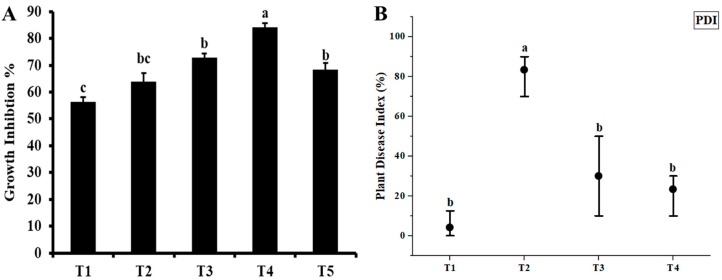
(**A**) In vitro antifungal activity and (**B**) in vivo potential of ZnO NPs to suppress Cercospora leaf spot disease at different concentrations: T1 = 0 mg/L (negative control); T2 = 0 mg/L, only pathogen (positive control); T3 = 900 mg/L + *C. canescens*; T4 = 1200 mg/L + *C. canescens*. The error bars represent the standard error of means. Different letters denote statistically significant differences between treatments as evaluated by the Tukey’s multiple range test at the *p* < 0.05 level.

**Figure 6 nanomaterials-15-00143-f006:**
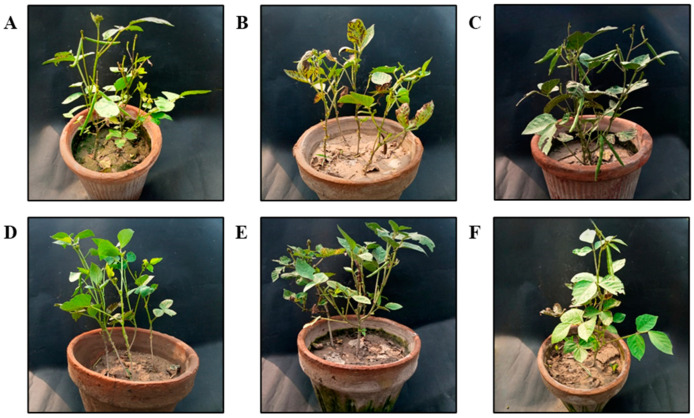
Effects of different ZnO NPs treatments on mung bean plants. (**A**) T1= 0 mg/L (negative control); (**B**) T2= only pathogen (positive control); (**C**) T3= 900 mg/L; (**D**) T4= 1200 mg/L; (**E**) T5= 900 mg/L + *C. canescens*; (**F**) T6= 1200 mg/L + *C. canescens*.

**Table 1 nanomaterials-15-00143-t001:** Treatments for evaluating the efficacy of green-synthesized ZnO NPs to augment mung bean plant growth and yield in vivo.

Serial Number	Treatments	Label
1	Negative control (without ZnO NPs treatment)	T1
2	Positive control (Only Pathogen)	T2
3	ZnO NPs solution (900 mg/L)	T3
4	ZnO NPs solution (1200 mg/L)	T4
5	ZnO NPs solution (900 mg/L) + *C. canescens*	T5
6	ZnO NPs solution (1200 mg/L) + *C. canescens*	T6

**Table 2 nanomaterials-15-00143-t002:** Effect of different ZnO NPs treatments on growth parameters of mung bean.

GO NPs Treatments (mg/L)	Shoot Length (cm)	Root Length (cm)	Number of Leaves (n)	Number of Pods per Plant (n)	No of Seeds per Pod (n)	Pod Size (cm)	Shoot Fresh Weight (g)	Shoot Dry Weight (g)	Root Fresh Weight (g)	Root Dry Weight (g)	Root Nodules (n)	Pod Fresh Weight (g)	Pod Dry Weight (g)
T1	34.13 ± 0.472 ^b^	14.03 ± 0.470 ^b^	22.0 ± 0.577 ^b^	8.00 ± 0.577 ^bc^	8.00 ± 0.577 ^abc^	7.00 ± 0.072 ^bc^	14.63 ± 0.504 ^ab^	4.30 ± 0.115 ^b^	1.36 ± 0.031 ^b^	0.48 ± 0.005 ^b^	12.6 ± 0.667 ^b^	1.30 ± 0.057 ^bc^	0.51 ± 0.022 ^bc^
T2	18.57 ± 0.857 ^d^	6.47 ± 0.857 ^d^	12.0 ± 1.154 ^c^	5.67 ± 0.577 ^d^	5.67 ± 0.333 ^c^	4.93 ± 0.233 ^d^	7.70 ± 0.550 ^d^	2.07 ± 0.059 ^c^	0.72 ± 0.046 ^d^	0.20 ± 0.012 ^d^	7.33 ± 0.67 ^d^	0.89 ± 0.080 ^e^	0.35 ± 0.032 ^e^
T3	37.31 ± 0.380 ^a^	17.21 ± 0.369 ^a^	25.0 ± 1.000 ^ab^	9.33 ± 0.333 ^ab^	10.0 ± 1.154 ^ab^	7.76 ± 0.145 ^ab^	15.6 ± 0.264 ^a^	6.05 ± 0.115 ^a^	1.51 ± 0.012 ^a^	0.57 ± 0.045 ^a^	14.0 ± 1.00 ^a^	1.49 ± 0.037 ^ab^	0.59 ± 0.014 ^ab^
T4	38.41 ± 0.240 ^a^	18.31 ± 0.251 ^a^	27.3 ± 0.333 ^a^	11.0 ± 0.577 ^a^	11.0 ± 1.000 ^a^	7.86 ± 0.185 ^a^	15.9 ± 0.152 ^a^	6.17 ± 0.088 ^a^	1.54 ± 0.011 ^a^	0.58 ± 0.044 ^a^	15.0 ± 0.57 ^a^	1.65 ± 0.032 ^a^	0.65 ± 0.012 ^a^
T5	31.47 ± 0.548 ^c^	11.37 ± 0.634 ^c^	22.0 ± 0.577 ^b^	6.67 ± 0.333 ^c^	7.0 ± 0.577 ^c^	6.40 ± 0.360 ^c^	12.52 ± 0.306 ^c^	4.00 ± 0.102 ^b^	1.20 ± 0.029 ^c^	0.42 ± 0.010 ^c^	9.33 ± 1.20 ^c^	1.09 ± 0.037 ^ce^	0.43 ± 0.014 ^ce^
T6	33.33 ± 0.448 ^bc^	13.23 ± 0.692 ^bc^	22.3 ± 0.667 ^b^	7.33 ± 0.667 ^bc^	8.67 ± 0.333 ^abc^	6.96 ± 0.070 ^bc^	14.0 ± 0.173 ^b^	4.17 ± 0.202 ^b^	1.35 ± 0.015 ^b^	0.47 ± 0.005 ^b^	11.6 ± 0.88 ^b^	1.29 ± 0.023 ^bcd^	0.51 ± 0.009 ^bcd^

Data represent mean ± standard error. Different letters denote statistically significant differences between treatments as evaluated by the ANOVA and Tukey’s multiple range test at the *p* < 0.05 level.

**Table 3 nanomaterials-15-00143-t003:** Effects of different concentrations of green-synthesized ZnO NPs on physiological parameters of mung bean plants.

Treatments (mg/L)	Chl *a*	Chl *b*	Total Chlorophyll	Carotenoids
T1	0.43145 ± 0.0184 ^b^	0.25473 ± 0.0062 ^b^	0.68619 ± 0.0216 ^b^	15.7123 ± 0.2312 ^ab^
T2	0.32140 ± 0.0241 ^c^	0.17106 ± 0.0052 ^c^	0.49247 ± 0.0266 ^c^	11.3025 ± 0.5230 ^c^
T3	0.54808 ± 0.0265 ^a^	0.31245 ± 0.0101 ^a^	0.86054 ± 0.0184 ^a^	16.0469 ± 0.8498 ^ab^
T4	0.55238 ± 0.0123 ^a^	0.32789 ± 0.0035 ^a^	0.88027 ± 0.0162 ^a^	17.4531 ± 1.1536 ^a^
T5	0.39641 ± 0.0117 ^bc^	0.22414 ± 0.0091 ^b^	0.62056 ± 0.0008 ^b^	13.4327 ± 1.1877 ^bc^
T6	0.41742 ± 0.0112 ^b^	0.23704 ± 0.0082 ^b^	0.65447 ± 0.0212 ^b^	14.0980 ± 0.8630 ^abc^

Data represent the mean ± standard error. Different letters denote statistically significant differences between treatments as evaluated by Tukey’s Multiple range test at the *p* < 0.05 level.

## Data Availability

The original contributions presented in the study are included in the article/Appendix A. Further inquiries can be directed to the corresponding authors.

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
