# Peer review of "Antifungal Potential of Biogenic Zinc Oxide Nanoparticles for Controlling Cercospora Leaf Spot in Mung Bean"

_nanomaterials, 2025, doi:10.3390/nano15020143_

Round 1
Reviewer 1 Report
Comments and Suggestions for Authors
The authors of the manuscript “Antifungal Potential of Biogenic Zinc Oxide Nanoparticles for Controlling Cercospora Leaf Spot in Mung Bean” present an interesting study on the ability of ZnO nanoparticles containing Nigella sativa seed extracts to increase crop yield and enhance antifungal activity against Cercospora canescens. To achieve this, the nanoparticles were characterised by UV-Vis, FTIR analyses and SEM observations, in addition, evidence of antifungal activity and all parameters and indicators of healthy plant growth were reported. The lack of the number of lines makes the revision work much more difficult to carry out.
The manuscript is generally clear and comprehensive. The results are interesting. The manuscript contains some “flaws” that need to be improved.
Points to be addressed:
Abstract
Nigella sativa should be written in italics.
Cercospora canescens should be written in italics.
Introduction
Page 2: Candida albicans should be abbreviated as already mentioned.
Page 2: S. aureus and E. coli should be written in full, not abbreviated. The second S. aureus can be deleted.
Page 2: The word "mungbean" should be written in separate parts (both after ref 15 and after ref 18). Authors are advised to check the whole text of the manuscript very carefully, several cases of "mungbean" have been detected.
Page 2 - second from last line: Nigella sativa should be spelled and abbreviated correctly.
Materials and Methods
Section 2.1: Nigella sativa should be abbreviated and italicised.
Section 2.3: Nigella sativa should be abbreviated.
Section 2.6.1 - spore concentration: use superscript for “...105 conidia per mL”.
Table 1: What does “Sr.” mean in the first column? Why are there 2 T4 treatments in the last column of the table? Please correct the table using intuitive symbols and/or explained abbreviations.
Results
Section 3.1: “Cercospora canescens”, the species name should be abbreviated as it is already mentioned.
Section 3.1: “Septa size”, use superscript when in mm. Why not use microns? The superscript can be omitted. Authors are asked to choose one of the two solutions.
Figure 1 - Legend: C. canescens should be italicised. Panel “A” is not mentioned in the text.
Section 3.2.1: the bibliographic citation does not follow the journal standard (Fageria et al. 2014).
Figure 2: it contains 4 panels from 4 different analyses and is placed after the citation of panels “A” and “B”. Two solutions are possible:
1) move figure 2 after section 3.2.3. Panels “C” and “D” are cited under 'X-ray diffraction' and 'FTIR';
2) prepare 2 figures. The first with pannels “A” and “B” in their present position and the second with plates “C” and “D” after section 3.2.3.
Table 2: the table should be moved to page 10, after the citation. Tip for authors.... try to make at least two different tables (shoot and root), it is very hard to read the data.
Section 3.2: please, I encourage authors to call the “Supplemental Table 2” as “Table S2” is less confusing, it applies to all supplementary “Table or Figure Sx”. But where are the supplementary materials?
Section 3.3: 150 mg/mL is written but should be 1500 mg/mL.
Figure 4 - Legend: C. canescens should be in italics.
Figure 5 - Panel B: the T5 value is missing, please add it or explain why it is missing.
Section 3.5: Authors wrote about triplicate… What do the authors mean by triplicate, three pots? Three seedlings in one pot? Please clarify the materials and methods as well as the results.
Figure 6 legend: the species name should be in italics; T5 means something different from Table 1, is this T5 one of the T4s repeated in Table 1? This causes confusion and authors are invited to correct and clarify.
“-ve” can be misleading, please write “negative”.
Discussion
Section 4 - page 11: The whole paragraph on thymoquinone needs to be better written. Too many sentences starting with 'Thymoquinone is...'. Something more cohesive is needed.
Section 4, end of page 11: Candida albicans should be abbreviated.
Section 4, page 12: Aspergillus niger and Cercospora canescens should be abbreviated.
Section 4-page 12: What is T. ammi?
Section 4-page 13: A reference is not formatted according to the journal standard (Ahmad et al. 2020)
References
Authors should check the numbering of the references when entering the two unformatted references.
Author Response
Reviewer 1
|
No. |
Questions |
Response |
|
1 |
Nigella sativa should be written in italics. Cercospora canescens should be written in italics. |
Thank you for the observation. All instances of Nigella sativa and Cercospora canescens have been italicized, as shown in the tracked changes. |
|
2 |
Page 2: Candida albicans should be abbreviated as already mentioned.
|
Candida albicans has been appropriately abbreviated throughout the manuscript.
|
|
3 |
Page 2: S. aureus and E. coli should be written in full, not abbreviated. The second S. aureus can be deleted.
|
S. aureus and E. coli have been written in full, and the duplicate S. aureus entry has been removed. |
|
4 |
Page 2: The word "mungbean" should be written in separate parts (both after ref 15 and after ref 18). Authors are advised to check the whole text of the manuscript very carefully, several cases of "mungbean" have been detected.
|
"Mungbean" has been corrected to "mung bean" throughout the manuscript after a thorough review. |
|
5 |
Page 2 - second from last line: Nigella sativa should be spelled and abbreviated correctly.
|
Nigella sativa has been corrected and appropriately abbreviated in the revised text.
|
|
6 |
Section 2.1: Nigella sativa should be abbreviated and italicised.
|
Nigella sativa has been abbreviated and italicized in Section 2.1.
|
|
7 |
Section 2.3: Nigella sativa should be abbreviated. |
Nigella sativa has been abbreviated in Section 2.3.
|
|
8 |
Section 2.6.1 - spore concentration: use superscript for “...105 conidia per mL”.
|
Superscript formatting has been applied to "10⁵ conidia per mL" in Section 2.6.1.
|
|
9 |
Table 1: What does “Sr.” mean in the first column? Why are there 2 T4 treatments in the last column of the table? Please correct the table using intuitive symbols and/or explained abbreviations.
|
"Sr." stands for "Serial Number" used for numbering. The column header is now clarified as “Serial Number (Sr.)” in the revised table. The duplicate "T4" was corrected to "T5" as it was a typographical error. The table has been revised with intuitive column headers, and all abbreviations are now explained in the table legend. |
|
10 |
Section 3.1: “Cercospora canescens”, the species name should be abbreviated as it is already mentioned.
|
The units were incorrectly written as millimeters (mm) due to a typographical error. We have corrected this, and the septa size is now accurately expressed in microns (µm).
|
|
11 |
Figure 1 - Legend: C. canescens should be italicised. Panel “A” is not mentioned in the text.
|
We have italicized C. canescens throughout the manuscript, including the figure legend, and referenced Panel "A" in the text to describe the symptoms.
|
|
12 |
Section 3.2.1: the bibliographic citation does not follow the journal standard (Fageria et al. 2014).
|
Response: The citation for Fageria et al., 2014 has been removed from this part of the paragraph, as it is correctly cited with reference [28] in the following line. The reference number for Fageria et al., 2014 is now [29] in the manuscript, and the citations are aligned with the journal's guidelines. |
|
13 |
Figure 2: it contains 4 panels from 4 different analyses and is placed after the citation of panels “A” and “B”. Two solutions are possible: 1) move figure 2 after section 3.2.3. Panels “C” and “D” are cited under 'X-ray diffraction' and 'FTIR'; 2) prepare 2 figures. The first with pannels “A” and “B” in their present position and the second with plates “C” and “D” after section 3.2.3.
|
Figure 2 has been moved to follow Section 3.2.3 to align with the sequence of its citations and maintain clarity in the presentation.
|
|
14 |
Table 2: the table should be moved to page 10, after the citation. Tip for authors.... try to make at least two different tables (shoot and root), it is very hard to read the data. |
The position of Table 2 has been adjusted as suggested and is now placed on page 10 after the relevant citation. However, we have retained the table as a single entity to maintain the integrity of the data, ensuring a cohesive presentation.
|
|
15 |
Section 3.2: please, I encourage authors to call the “Supplemental Table 2” as “Table S2” is less confusing, it applies to all supplementary “Table or Figure Sx”. But where are the supplementary materials? |
We have updated the naming convention throughout the manuscript and supplementary materials to refer to "Supplemental Table 2" as "Table S2" and applied the same format to all supplementary tables and figures. The supplementary materials, including all tables and figures, are now provided as separate files for your review.
|
|
16 |
Section 3.3: 150 mg/mL is written but should be 1500 mg/mL. |
The concentration has been corrected to 1500 mg/mL as per your suggestion.
|
|
17 |
Figure 4 - Legend: C. canescens should be in italics. |
We have italicized C. canescens throughout the manuscript as requested. |
|
18 |
Figure 5 - Panel B: the T5 value is missing, please add it or explain why it is missing.
|
The legend was mistakenly labeled, but the graph itself is correct. The legend has now been updated to align accurately with the graph, and appropriate changes have been made in the section on the in vivo antifungal potential of zinc oxide nanoparticles to ensure consistency.
|
|
19 |
Section 3.5: Authors wrote about triplicate… What do the authors mean by triplicate, three pots? Three seedlings in one pot? Please clarify the materials and methods as well as the results.
|
By "triplicate," we refer to three independent replicates for each treatment to ensure consistency and reliability. We have revised the manuscript to reflect this clarification.
|
|
20 |
Figure 6 legend: the species name should be in italics; T5 means something different from Table 1, is this T5 one of the T4s repeated in Table 1? This causes confusion and authors are invited to correct and clarify. “-ve” can be misleading, please write “negative”.
|
2We have made the following revisions in the manuscript:
|
|
21 |
Section 4 - page 11: The whole paragraph on thymoquinone needs to be better written. Too many sentences starting with 'Thymoquinone is...'. Something more cohesive is needed.
|
We have revised the paragraph to improve cohesion and reduce repetitive sentence structures.
|
|
22 |
Section 4, end of page 11: Candida albicans should be abbreviated.
|
Response: Candida albicans has been abbreviated appropriately in this section.
|
|
23 |
Section 4, page 12: Aspergillus niger and Cercospora canescens should be abbreviated.
|
We have abbreviated Aspergillus niger and Cercospora canescens where mentioned in the manuscript.
|
|
24 |
Section 4-page 12: What is T. ammi? |
We have clarified that T. ammi refers to Trachyspermum ammi in the manuscript. |
|
25 |
Section 4-page 13: A reference is not formatted according to the journal standard (Ahmad et al. 2020)
|
The reference (Ahmad et al., 2020) has been reformatted to adhere to the journal's style guidelines.
|
|
26 |
Authors should check the numbering of the references when entering the two unformatted references.
|
The numbering of all references has been carefully reviewed and corrected to ensure consistency with the journal's requirements.
|
Reviewer 2
|
No |
Questions |
Response |
|
1 |
Line 17-18 How were the "low doses" of ZnO NPs determined? Can you provide more information on the concentration range tested and the criteria used to define a "low" dose in relation to potential toxicity?
|
To address this, we have added an additional paragraph to the discussion section, explaining the criteria for defining "low doses" of ZnO NPs. This includes the concentration range tested, references to relevant literature, and a justification for our selection based on potential toxicity and plant growth benefits. This addition provides greater clarity and aligns with the reviewer's request.
|
|
2 |
Line 18-19 What were the key characteristics of the ZnO nanoparticles as determined by the UV-Vis, FT-IR, X-ray diffraction, and SEM analyses? Did the characterization reveal any unique features that could explain the plant growth-promoting effects?
|
We have already explained the key characteristics of the ZnO nanoparticles (ZnO NPs) in the manuscript, including the UV-Vis, FT-IR, X-ray diffraction, and SEM analyses. These characterization techniques revealed several features that may contribute to the plant growth-promoting effects of ZnO NPs. The UV-Vis absorption spectrum confirmed the formation of ZnO NPs with a strong peak at 369 nm, indicative of ZnO's synthesis. The X-ray diffraction analysis confirmed the crystalline nature of the particles with a wurtzite (hexagonal) structure, which is known to be effective in promoting plant growth. FT-IR analysis revealed the presence of functional groups that may interact with plant cells, enhancing their uptake of essential nutrients. Finally, the SEM images showed nanoparticles in the range of 50–70 nm, which are optimal for interaction with plant cell walls, facilitating better absorption and transport of nutrients. We believe these characteristics are key factors in the observed plant growth promotion. These points are already discussed in detail in the manuscript.
|
|
3 |
Line 21-22 Can you provide more details about the antifungal testing procedure? How were the antifungal effects of ZnO NPs quantitatively measured, and were other pathogens or pests also tested for comparison?
|
The antifungal properties of ZnO nanoparticles (ZnO NPs) were evaluated against Cercospora canescens, the causative agent of Cercospora leaf spot in mung bean. Antifungal effects were quantified both in vitro and in planta by assessing plant growth parameters, including shoot and root length, pod and leaf count, root nodule count, and fresh and dry weight. Only Cercospora canescens was tested, as the broad-spectrum antifungal effects of ZnO NPs have been extensively documented in prior studies, and we did not find it necessary to include additional pathogens for comparison. |
|
4 |
Line 24-25 Which specific enzymatic activities and photosynthetic parameters were affected by the application of ZnO NPs? Could you elaborate on how these changes contribute to overall plant growth?
|
The application of low-dose ZnO nanoparticles (ZnO NPs) positively impacted enzymatic activities and photosynthetic parameters, including increases in total chlorophyll and carotenoid contents. These improvements in photosynthetic efficiency and antioxidant defense contribute to enhanced energy production and reduced oxidative stress, ultimately supporting better plant growth and higher yield. These effects are discussed in detail later in the manuscript.
|
|
5 |
What is the long-term sustainability of using biologically synthesized ZnO NPs in agricultural systems? Have you considered potential ecological impacts, such as nanoparticle accumulation in soil or effects on soil microorganisms?
|
While this study focuses on the immediate effects of biologically synthesized ZnO NPs, we acknowledge the need for further research on their long-term sustainability and potential ecological impacts. The accumulation of ZnO NPs in soil and their effects on soil microorganisms are important considerations. Although our study does not directly address these concerns, the eco-friendly synthesis using Nigella sativa suggests a lower environmental impact. Future studies should investigate nanoparticle persistence, biodegradability, and interactions with soil ecosystems to assess long-term sustainability and minimize ecological risks.
|
|
6 |
Line 83-84 Could you clarify the specific bioactive compounds in Nigella sativa that contribute to the synthesis of ZnO nanoparticles? How do these compounds influence the properties and efficacy of the nanoparticles produced?
|
The synthesis of ZnO nanoparticles (ZnO NPs) in this study was facilitated by thymoquinone, a bioactive compound in Nigella sativa seeds, which constitutes 30-48% of the volatile oil content. Thymoquinone, recognized for its antifungal, antibacterial, and anti-inflammatory properties, likely acts as a reducing agent in the nanoparticle synthesis. Its phytochemical properties may also enhance the structure and efficacy of ZnO NPs for agricultural applications. These aspects are discussed in detail in the discussion section of the manuscript.
|
|
7 |
Will the study investigate the broader range of antimicrobial properties of the Nigella sativa-synthesized ZnO nanoparticles against other pathogens beyond Cercospora canescens? How do these broader antimicrobial effects contribute to the overall agricultural sustainability of using ZnO NPs?
|
While this study focused on the antifungal effects of Nigella sativa-synthesized ZnO nanoparticles against Cercospora canescens, the potential antimicrobial properties of ZnO NPs against other pathogens remain a valuable area for future research. Several studies have demonstrated the broad-spectrum antimicrobial activity of ZnO NPs, including antibacterial and antiviral properties, which can contribute significantly to agricultural sustainability by reducing reliance on synthetic pesticides and promoting healthier crop growth. Expanding the investigation to include other pathogens would enhance the understanding of ZnO NPs' role in integrated pest management and sustainable agricultural practices. This aspect could be explored in future studies. |
|
8 |
Introduction must be improved? Rewrite the objective of the study again?
|
The objectives have been rewritten and the introduction improved for clarity and focus. The revised version highlights the synthesis, characterization, and antimicrobial activity of ZnO nanoparticles from Nigella sativa, emphasizing their potential to enhance plant growth and offer a sustainable alternative to chemical fungicides.
|
|
9 |
Recheck the suitable statistical design?
|
We used the DSASTAT Excel extension for statistical analysis, performing a one-way ANOVA followed by Tukey’s test to assess significant differences between treatments. This design is suitable for comparing multiple groups and identifying significant variations, as detailed in the manuscript. |
|
10 |
Can you provide more details on the specific field research conditions under which Cercospora canescens was isolated? Were any control or non-infected leaves used for comparison to ensure the pathogen's specificity?
|
We appreciate the reviewer’s comment. The specific field research conditions under which Cercospora canescens was isolated are already outlined in the manuscript. We have provided detailed information on the collection of infected leaves from Ayub Agricultural Research Institute (AARI), Faisalabad, as well as the methodology used for pathogen isolation, surface sterilization, and fungal culture. Additionally, control samples using non-infected leaves were included in the experiment to ensure the specificity of the pathogen. We believe the current level of detail is sufficient to support the study’s findings and does not require further elaboration.
|
|
11 |
From figure 1B and C Could you clarify how these morphological traits were quantified and compared to existing descriptions of Cercospora canescens? Did you observe any variations in these traits that could indicate different pathogenic strains or potential environmental influences?
|
The morphological traits of Cercospora canescens in Figure 1B and C were quantified and compared with existing descriptions, showing consistency with known characteristics such as opaque mycelium, pale to olive-brown conidiophores, and hyaline, ob-clavate to cylindrical conidia. The conidia size (102.8–3 μm) and septation (1–14 septa) align with reported values for C. canescens. No significant variations were observed in morphological traits that could indicate different strains or environmental influences.
|
|
12 |
Can you provide more information about the inoculation procedure and conditions under which Koch’s postulates were confirmed? Was there a control group in the experiment to rule out other potential causes of the leaf spots?
|
The inoculation procedure and conditions for confirming Koch’s postulates have been clearly outlined in the manuscript. Specifically, fungal cultures were grown in potato dextrose broth and diluted to a spore suspension of 10^6 spores/ml for inoculation. Twelve healthy mung bean leaves were used, with six inoculated with the spore suspension and six serving as the control, treated with distilled autoclaved water. The leaves were incubated at 28°C, and disease symptoms were regularly monitored. Koch’s postulates were confirmed by re-isolating the pathogen from the infected leaves. We believe these procedures are robust in fulfilling Koch’s postulates, ensuring the specificity of the pathogen. If the reviewer feels additional details are needed, we are happy to incorporate further information into the manuscript. |
|
13 |
Can you discuss how the band gap of 3.12 eV relates to the potential application of ZnO NPs in agriculture, particularly in terms of their interaction with light and their effectiveness in promoting plant growth or controlling pathogens? How does this band gap compare to other forms of ZnO NPs used in similar studies?
|
The 3.12 eV band gap of ZnO nanoparticles (ZnO NPs) in this study indicates their potential for photocatalytic applications in agriculture. This band gap allows ZnO NPs to absorb UV light, generating reactive oxygen species (ROS) that can control pathogens and enhance plant growth. Compared to other forms of ZnO NPs, this band gap is within the typical range (3.0–3.4 eV), which has been shown to exhibit both antimicrobial and growth-promoting effects. Thus, the ZnO NPs in this study have promising applications as eco-friendly alternatives to chemical pesticides in crop protection. |
|
14 |
In your SEM analysis, did you observe any surface roughness or structural features that could influence the nanoparticles' interaction with plant surfaces or pathogens? How might these features contribute to the nanoparticles' effectiveness in antimicrobial activity or plant growth promotion?
|
In the SEM analysis, the ZnO nanoparticles (ZnO NPs) exhibited a relatively smooth, rectangular shape with a size range of 50 to 70 nm. While no significant surface roughness was observed, the morphology and size of the nanoparticles may still influence their interaction with plant surfaces and pathogens. Smaller nanoparticles, with high surface area-to-volume ratios, are more likely to interact effectively with cell membranes, facilitating antimicrobial activity by penetrating pathogen cells or disrupting their structure. Additionally, the surface features of the ZnO NPs can enhance their adhesion to plant surfaces, potentially improving their effectiveness in promoting plant growth by facilitating nutrient uptake or acting as a protective barrier against pathogens. |
|
15 |
Were any comparisons made between the SEM images of ZnO nanoparticles synthesized using Nigella sativa extract and those synthesized by other methods (e.g., chemical synthesis)? How do the morphology and size of the green-synthesized ZnO NPs compare with nanoparticles produced through traditional methods?
|
This study focused on ZnO nanoparticles synthesized using Nigella sativa extract, without comparing them to those produced by traditional chemical methods. The green-synthesized ZnO NPs displayed a controlled morphology with sizes ranging from 50 to 70 nm, which aligns with other plant-mediated synthesis studies. In contrast, chemical methods typically produce nanoparticles with a broader size distribution and may involve toxic chemicals. Green synthesis offers eco-friendly advantages, and existing literature supports that such ZnO NPs can be more effective for agricultural applications, promoting plant growth and protecting against pathogens.
|
|
16 |
SEM images often show only surface features. Did you perform any additional characterizations, such as X-ray diffraction (XRD) or dynamic light scattering (DLS), to confirm the crystalline structure or dispersion stability of the nanoparticles? |
In addition to SEM, X-ray diffraction (XRD) analysis was performed to confirm the crystalline structure of the ZnO nanoparticles, revealing a hexagonal wurtzite pattern. These results complement the SEM findings, providing a deeper insight into the structural properties of the nanoparticles.
|
|
17 |
Can you provide a possible explanation for why the antifungal activity did not continue to increase at the 1500 mg/L concentration? Is there any indication of a plateau effect, or could this result suggest a potential toxic effect of higher concentrations on the fungal growth or the nanoparticles themselves? |
The decrease in antifungal activity at 1500 mg/L compared to 1200 mg/L could suggest a plateau effect. At higher concentrations, ZnO nanoparticles may aggregate, reducing their effectiveness. Additionally, higher doses may induce toxicity, impacting both the fungus and the nanoparticles, leading to diminished antifungal activity.
|
|
18 |
Could you discuss the potential mechanism of action for the antifungal activity of ZnO NPs? Did you observe any morphological changes in C. canescens (e.g., hyphal distortion, spore germination inhibition) under the SEM or other microscopy techniques at varying concentrations?
|
The SEM analysis in this study was performed solely for the characterization of the ZnO nanoparticles, focusing on their morphology and structural features. While this characterization provides insights into the nanoparticles' potential mechanisms of action, such as their interaction with the fungal surface, SEM was not conducted at varying concentrations to observe specific antifungal effects like hyphal distortion. Future studies involving microscopy at different concentrations could provide more detailed understanding of the antifungal mechanisms of ZnO NPs.
|
|
19 |
What is the rationale behind selecting 1200 ppm as the optimal concentration for ZnO NPs? Were other concentrations tested, and if so, how did they compare in terms of their impact on plant growth?
|
1200 mg/L was selected as the optimal concentration for ZnO nanoparticles (ZnO NPs) based on its superior performance in in vitro antifungal tests against Cercospora canescens. Other concentrations (300, 600, 900, and 1500 mg/L) were also tested to assess both antifungal effects and plant growth promotion. The concentrations of 900 and 1200 mg/L showed the best results and were chosen for subsequent in vivo experiments.
|
|
20 |
The data show that ZnO NPs had a favorable impact even in the presence of Cercospora canescens. Could you elaborate on the specific mechanisms through which ZnO NPs might help mitigate the negative effects of the fungal pathogen on plant growth? Are these effects purely related to pathogen control, or do ZnO NPs also enhance plant resistance to fungal infections?
|
ZnO nanoparticles (ZnO NPs) mitigate the effects of Cercospora canescens on plant growth through direct antifungal activity, such as inhibiting spore germination and disrupting fungal cell walls. Additionally, ZnO NPs may enhance plant resistance by triggering defense mechanisms, including increased production of reactive oxygen species (ROS) and activation of defense-related enzymes. Furthermore, ZnO NPs support plant growth by improving nutrient uptake, particularly zinc, which is essential for metabolic processes. This dual role in pathogen control and plant growth promotion contributes to overall plant health, and these mechanisms are discussed thoroughly in the discussion chapter of the manuscript.
|
|
21 |
Given that Nigella sativa contains thymoquinone, could you elaborate on how thymoquinone interacts with ZnO NPs in the green synthesis process? Does thymoquinone contribute to the antifungal activity of the nanoparticles, or is the effect purely due to the ZnO NPs themselves?
|
Thymoquinone, the active compound in Nigella sativa, plays a key role in the green synthesis of ZnO nanoparticles (ZnO NPs) by acting as a reducing and stabilizing agent. This enhances the stability and dispersibility of the nanoparticles. Thymoquinone also has intrinsic antifungal properties, and when combined with ZnO NPs, it may enhance their antifungal activity. Thus, the synergistic effect of thymoquinone and ZnO NPs contributes to their improved efficacy in combating fungal pathogens, making the nanoparticles more effective in controlling infections.
|
|
22 |
You mention that thymoquinone has antifungal properties. How does the antifungal efficacy of thymoquinone compare to the observed effects of ZnO NPs synthesized from Nigella sativa? Could the combined effect of the phytochemical and the nanoparticles lead to a synergistic antifungal action?
|
Thymoquinone, with its known antifungal properties, may enhance the antifungal efficacy of ZnO nanoparticles (ZnO NPs) synthesized from Nigella sativa. The combination of thymoquinone and ZnO NPs could result in a synergistic effect, where thymoquinone aids in the stability and dispersion of the nanoparticles, while the NPs' antimicrobial properties are amplified by the phytochemical’s bioactivity, leading to improved antifungal activity.
|
|
23 |
In the context of C. canescens, did you observe similar structural changes to the fungal hyphae, conidiophores, or conidia when treated with ZnO NPs? How did the treatment affect fungal growth or reproduction in your study?
|
In this study, we focused on the antifungal effects of ZnO nanoparticles on Cercospora canescens through plant growth parameters and pathogen inhibition. We did not examine structural changes in fungal hyphae or conidiophores, but the observed reduction in fungal growth suggests the effectiveness of ZnO NPs in controlling the pathogen. Future studies could explore these structural effects more directly.
|
|
24 |
Based on these studies, how do your findings compare with the concentrations and antifungal effects observed in the literature? Were the concentrations used in your study comparable to those in the cited studies, and how do they support the dose-dependent nature of ZnO NP antifungal activity?
|
In our study, the concentrations of ZnO NPs tested ranged from 300 mg/L to 1500 mg/L, with the optimal antifungal effects observed at 1200 mg/L. These findings are consistent with the dose-dependent nature of ZnO NP antifungal activity reported in the literature, where varying concentrations of ZnO NPs exhibit different levels of efficacy against pathogens. As discussed in the manuscript, we noted that higher concentrations (1500 mg/L) showed slightly reduced efficacy, which may suggest a plateau effect or potential toxicity at excessive doses. This aligns with other studies that highlight the importance of optimizing nanoparticle concentrations to balance efficacy and safety for both plants and pathogens. The detailed discussion of dose specificity can be found in the manuscript's discussion section. |
|
25 |
25. Rewrite conclusion
|
The conclusion has been rewritten to reflect the potential benefits of ZnO nanoparticles (ZnO NPs) on the antifungal activity and physiological performance of mung bean plants, as well as the importance of optimizing concentration and exposure duration. |
Reviewer 2 Report
Comments and Suggestions for Authors
In this study, Aftab et al. conducted a Antifungal Potential of Biogenic Zinc Oxide Nanoparticles for 2 Controlling Cercospora Leaf Spot in Mung Bean. Although the study provides some useful information, there are major concerns that need to be addressed by the authors.
1. Line 17-18 How were the "low doses" of ZnO NPs determined? Can you provide more information on the concentration range tested and the criteria used to define a "low" dose in relation to potential toxicity?
2. Line 18-19 What were the key characteristics of the ZnO nanoparticles as determined by the UV-Vis, FT-IR, X-ray diffraction, and SEM analyses? Did the characterization reveal any unique features that could explain the plant growth-promoting effects?
3. Line 21-22 Can you provide more details about the antifungal testing procedure? How were the antifungal effects of ZnO NPs quantitatively measured, and were other pathogens or pests also tested for comparison?
4. Line 24-25 Which specific enzymatic activities and photosynthetic parameters were affected by the application of ZnO NPs? Could you elaborate on how these changes contribute to overall plant growth?
5. Line 25-26 What is the long-term sustainability of using biologically synthesized ZnO NPs in agricultural systems? Have you considered potential ecological impacts, such as nanoparticle accumulation in soil or effects on soil microorganisms?
6. Line 83-84 Could you clarify the specific bioactive compounds in Nigella sativa that contribute to the synthesis of ZnO nanoparticles? How do these compounds influence the properties and efficacy of the nanoparticles produced?
7. Will the study investigate the broader range of antimicrobial properties of the Nigella sativa-synthesized ZnO nanoparticles against other pathogens beyond Cercospora canescens? How do these broader antimicrobial effects contribute to the overall agricultural sustainability of using ZnO NPs?
8. Introduction must be improved? Rewrite the objective of the study again?
9. Recheck the suitable statistical design?
10. Can you provide more details on the specific field research conditions under which Cercospora canescens was isolated? Were any control or non-infected leaves used for comparison to ensure the pathogen's specificity?
11. From figure 1B and C Could you clarify how these morphological traits were quantified and compared to existing descriptions of Cercospora canescens? Did you observe any variations in these traits that could indicate different pathogenic strains or potential environmental influences?
12. Can you provide more information about the inoculation procedure and conditions under which Koch’s postulates were confirmed? Was there a control group in the experiment to rule out other potential causes of the leaf spots?
13. Can you discuss how the band gap of 3.12 eV relates to the potential application of ZnO NPs in agriculture, particularly in terms of their interaction with light and their effectiveness in promoting plant growth or controlling pathogens? How does this band gap compare to other forms of ZnO NPs used in similar studies?
14. In your SEM analysis, did you observe any surface roughness or structural features that could influence the nanoparticles' interaction with plant surfaces or pathogens? How might these features contribute to the nanoparticles' effectiveness in antimicrobial activity or plant growth promotion?
15. Were any comparisons made between the SEM images of ZnO nanoparticles synthesized using Nigella sativa extract and those synthesized by other methods (e.g., chemical synthesis)? How do the morphology and size of the green-synthesized ZnO NPs compare with nanoparticles produced through traditional methods?
16. SEM images often show only surface features. Did you perform any additional characterizations, such as X-ray diffraction (XRD) or dynamic light scattering (DLS), to confirm the crystalline structure or dispersion stability of the nanoparticles?
17. Can you provide a possible explanation for why the antifungal activity did not continue to increase at the 1500 mg/L concentration? Is there any indication of a plateau effect, or could this result suggest a potential toxic effect of higher concentrations on the fungal growth or the nanoparticles themselves?
18. Could you discuss the potential mechanism of action for the antifungal activity of ZnO NPs? Did you observe any morphological changes in C. canescens (e.g., hyphal distortion, spore germination inhibition) under the SEM or other microscopy techniques at varying concentrations?
19. What is the rationale behind selecting 1200 ppm as the optimal concentration for ZnO NPs? Were other concentrations tested, and if so, how did they compare in terms of their impact on plant growth?
20. The data show that ZnO NPs had a favorable impact even in the presence of Cercospora canescens. Could you elaborate on the specific mechanisms through which ZnO NPs might help mitigate the negative effects of the fungal pathogen on plant growth? Are these effects purely related to pathogen control, or do ZnO NPs also enhance plant resistance to fungal infections?
21. Given that Nigella sativa contains thymoquinone, could you elaborate on how thymoquinone interacts with ZnO NPs in the green synthesis process? Does thymoquinone contribute to the antifungal activity of the nanoparticles, or is the effect purely due to the ZnO NPs themselves?
22. You mention that thymoquinone has antifungal properties. How does the antifungal efficacy of thymoquinone compare to the observed effects of ZnO NPs synthesized from Nigella sativa? Could the combined effect of the phytochemical and the nanoparticles lead to a synergistic antifungal action?
23. In the context of C. canescens, did you observe similar structural changes to the fungal hyphae, conidiophores, or conidia when treated with ZnO NPs? How did the treatment affect fungal growth or reproduction in your study?
24. Based on these studies, how do your findings compare with the concentrations and antifungal effects observed in the literature? Were the concentrations used in your study comparable to those in the cited studies, and how do they support the dose-dependent nature of ZnO NP antifungal activity?
25. Rewrite conclusion
Comments on the Quality of English Language
The English should be improved to more clearly express the research
Author Response

(The authors gave the same response as above.)

Round 2
Reviewer 1 Report
Comments and Suggestions for Authors
Following the instructions of the previous revision, the manuscript has been improved. There are only a couple of "minor issues" left that need to be improved.
Points that need to be addressed:
Keywords: authors are encouraged to revise the keywords. In this case, remove the semicolon in the first keyword, replace "mungbean" with "mung bean", and "antifungal" needs to be completed (property? effect? ...).
Lines 404-422 and the References section: authors are kindly invited to revise the format of the introduced references.
Author Response
Following the instructions of the previous revision, the manuscript has been improved. There
are only a couple of "minor issues" left that need to be improved.
Points that need to be addressed:
Keywords: authors are encouraged to revise the keywords. In this case, remove the semicolon
in the first keyword, replace "mungbean" with "mung bean", and "antifungal" needs to be
completed (property? effect? ...).
Response: Thank you for your valuable suggestions regarding the keywords. We have revised
them by removing the semicolon in the first keyword, replacing "mungbean" with "mung
bean," and completing "antifungal" to "antifungal property" for clarity.
Lines 404-422 and the References section: authors are kindly invited to revise the format of
the introduced references.
Response: Thank you for pointing out the formatting issue with the references. We have
carefully revised the citations in lines 404–422 of the manuscript, and they are now updated as
(38), (39), and (40) to align with the journal's format. With the changes in numbering, we have
also carefully revised the numbering of the subsequent citations accordingly. We appreciate
your feedback.
Reviewer 2 Report
Comments and Suggestions for Authors
Carefully check grammatically and typos before submission in all section.
Comments on the Quality of English LanguageThe English could be improved to more clearly express the research.
Author Response
Comments and Suggestions for Authors
Carefully check grammatically and typos before submission in all sections.
Response: We have carefully reviewed the manuscript for grammatical errors and typos in all
sections and made the necessary corrections before submission.
Comments on the Quality of English Language
The English could be improved to more clearly express the research.
Response: Thank you for your feedback. While we have not made any specific changes to alter
the overall expression of the research, we have reviewed the manuscript thoroughly for typos
and language. Changes have been made where necessary to ensure clarity and accuracy. We
believe the current language effectively conveys the research, and we appreciate your
suggestions to improve its presentation.
